# Influence of previous Zika virus infection on acute dengue episode

**Cassia F. Estofolete**[1], **Alice F. Versiani**[1,2], **Fernanda S. Dourado**[1], **Bruno H. G. A. Milhim**[1], **Carolina C. Pacca**[1], **Gislaine C. D. Silva**[1], **Nathalia Zini**[1], **Barbara F. dos Santos**[1], **Flora A. Gandolfi**[1], **Natalia F. B. Mistrão**[1], **Pedro H. C. Garcia**[1], **Rodrigo S. Rocha**[1], **Lee Gehrke**[3,4], **Irene Bosch**[3], **Rafael E. Marques**[5], **Mauro M. Teixeira**[6], **Flavio G. da Fonseca**[7,8], **Nikos Vasilakis**[2,9,10,11,12], **Maurício L. Nogueira** [1,2]*

1 Laboratório de Pesquisas em Virologia (LPV), Faculdade de Medicina de São José do Rio Preto (FAMERP); São José do Rio Preto, Sao Paulo, Brazil, 2 Department of Pathology, University of Texas Medical Branch; Galveston, Texas, United States of America, 3 Institute for Medical Engineering and Science, Massachusetts Institute of Technology; Cambridge, Massachusetts, United States of America, 4 Department of Microbiology, Harvard Medical School; Boston, Massachusetts, United States of America, 5 Brazilian Biosciences National Laboratory (LNBio), Brazilian Center for Research in Energy and Materials (CNPEM); Campinas, Sao Paulo, Brazil, 6 Department of Biochemistry and Immunology, Universidade Federal de Minas Gerais; Belo Horizonte, Minas Gerais, Brazil, 7 Laboratório de Virologia Básica e Aplicada, Departamento de Microbiologia, Instituto de Ciências Biológicas, Universidade Federal de Minas Gerais; Belo Horizonte, Minas Gerais, Brazil, 8 Centro de Tecnoogia em Vacinas da UFMG, Universidade Federal de Minas Gerais; Belo Horizonte, Minas Gerais, Brazil, 9 Center for Vector-Borne and Zoonotic Diseases, University of Texas Medical Branch; Galveston, Texas, United States of America, 10 Center for Biodefense and Emerging Infectious Diseases, University of Texas Medical Branch; Galveston, Texas, United States of America, 11 Center for Tropical Diseases, University of Texas Medical Branch; Galveston, Texas, United States of America, 12 Institute for Human Infection and Immunity, University of Texas Medical Branch; Galveston, Texas, United States of America

* mauricio.nogueira@edu.famerp.br

## Abstract

### Background

The co-circulation of flaviviruses in tropical regions has led to the hypothesis that immunity generated by a previous dengue infection could promote severe disease outcomes in subsequent infections by heterologous serotypes. This study investigated the influence of antibodies generated by previous Zika infection on the clinical outcomes of dengue infection.

### Methodology/Principal findings

We enrolled 1,043 laboratory confirmed dengue patients and investigated their prior infection to Zika or dengue. Severe forms of dengue disease were more frequent in patients with previous Zika infection, but not in those previously exposed to dengue.

### Conclusions/Significance

Our findings suggest that previous Zika infection may represent a risk factor for subsequent severe dengue disease, but we did not find evidence of antibody-dependent enhancement (higher viral titer or pro-inflammatory cytokine overexpression) contributing to exacerbation of the subsequent dengue infection.

**Data Availability Statement:** The data is available and was uploaded as files named S1 and S2 Data.

**Funding:** This study was supported by the São Paulo Research Foundation (FAPESP) via grant

2022/03645-1 for MLN, 2018/17647-0 for AFV, and 2022/09229-0 for CFE, by INCT Dengue Program grant 465425/2014-3 (MLN), and by INCT Viral Genomic Surveillance and One Health grant 405786/2022-0. MLN is Brazilian National Council for Scientific and Technological Development (CNPq) Research Fellows. MLN and NV are partly funded by the Centers for Research in Emerging Infectious Diseases (CREID), "The Coordinating Research on Emerging Arboviral Threats Encompassing the Neotropics (CREATE-NEO)" grant U01AI151807 (to NV) by the National Institutes of Health (NIH/USA). The funders had no role in study design, data collection and analysis, decision to publish, or preparation of the manuscript.

**Competing interests:** The authors have declared that no competing interests exist.

## Author summary

Although dengue is a disease known for years in the world and has been affecting several continents, some aspects remain unclear. One of them is about the possible factors that may influence the development of severe forms of the disease. Much has been discussed about the influence of a previous dengue episode, but the global spreading of other flaviviruses to areas where dengue was already circulating has aroused interest regarding a role like the other dengue serotypes. In this sense, the recent spreading of the Zika virus has become a factor of interest. In this study, the prior Zika virus infection was associated with a higher frequency of more severe forms in subsequent dengue. Preliminary findings did not suggest that the mechanism is the same one triggered in secondary dengue, known as antibody-dependent enhancement. These findings are a stimulus to develop further research that can understand the potential mechanisms involved in the pathogenesis.

## Introduction

In recent years the arbovirus infections have become a significant health concern worldwide. The hyperendemicity of dengue virus (DENV) in tropical regions highlights its impact on human health and the global economy [1]. Since its reintroduction in Brazil in the mid-1980's, the co-circulation of all four serotypes have been observed throughout the country leading to increased disease severity [2, 3].

Infection by any of the four DENV serotypes may cause an acute febrile illness and lead to severe and potentially fatal clinical outcomes [reviewed in [4]]. While most patients recover from a self-limited illness, a small number progress to severe dengue disease (SDD), with a mortality rate of ca. 1% [5]. The pathogenesis of SDD is quite complex and is a multifactorial process mostly observed in secondary heterologous DENV infections. Potential outcomes of the secondary heterologous infections include an intense inflammatory response culminating in increased viral titers and exacerbated immune activation, a process known as antibody-dependent enhancement (ADE) [6–9]. ADE occurs when non-neutralizing antibodies from a previous DENV infection bind to DENV in a subsequent heterotypic infection but cannot neutralize the virus. This results in the binding and entry of antibody-virus complexes to the Fcγ receptors (FcγR) on circulating monocytes, thus exacerbating clinical presentations manifested by hemodynamic changes, increased viremia, and proinflammatory cytokine profiles [as reviewed in [10–12]].

In 2019 South America experienced an unprecedent dengue epidemic with 3,139,335 reported dengue cases, exceeding the 2015–2016 DENV epidemic by 30% [13], of which 28,169 (0.9%) cases were classified as SDD, with 1,538 deaths (0.049%). Interestingly, this epidemic occurred four years after the emergence of Zika virus (ZIKV) on the continent. Indeed, ZIKV shares approximately 58% genetic similarity with DENV [14], which raises two questions: i) the potential of cross-reactivity; and ii) whether prior ZIKV immunity will impact the immune response and clinical outcomes of subsequent DENV infections [15, 16]. Therefore, understanding the role of pre-existing immunity to genetically and antigenically similar pathogens may lead to effective clinical protocols to treat or control exacerbated forms of the disease [15].

Both ZIKV and DENV trigger similar serological responses in hosts. IgM antibodies are detected within the first week of infection and used as a diagnostic tool of acute infection,

whereas detection of IgG signifies seroconversion and used as a marker of past exposure [17, 18]. Once the envelope protein of ZIKV and DENV share a 53.9% amino acid identity [19], and is an important target for neutralizing antibodies [20], this raises the spectra of cross-reactivity in diagnostic tests, thus rendering accurate diagnosis a challenge [19, 21].

In addition, several studies *in vitro* have shown enhancement of ZIKV infection in the presence of DENV antibodies [15, 22–25], and outcomes from *in vivo* studies have been inconclusive [26–32]. A previous ZIKV infection may stimulate non-specific adaptive immunity that eventually culminate in cytokine storm and inflammatory responses leading to severe forms of dengue. Namely, Bardina et al.[26] observed severe outcomes in mice treated by DENV or WNV-convalescent plasma and infected by ZIKV, including higher ZIKV mortality in groups that have experienced previous DENV infection, compared to the control group (no prior DENV infection). In contrast, in a human cohort in Bahia, Brazil, Rodriguez-Barraquer et al. [27] showed that pre-existing immunity to dengue was protective to subsequent ZIKV infection, a conclusion confirmed in a human cohort in Sao Jose do Rio Preto, Sao Paulo [33] and non-human primates, in which Terzian et al. [33] and Pantoja et al. [27], respectively, did not observe increased severity of ZIKV infection in humans or NHPs that had been exposed to prior DENV infection. Thus, the role of prior DENV infection influencing the clinical outcome of subsequent ZIKV infection remains poorly understood.

Conversely, the role of previous ZIKV infection on subsequent dengue is unclear. Critically, a recent study in a pediatric cohort in Nicaragua [34] reported that the risk of symptomatic or SDD was higher in children with one previous ZIKV or DENV infection, but, interestingly, not in children exposed to multiple DENV infections, suggesting that prior ZIKV infection is a modulator of symptomatic and severe disease clinical outcomes. In 2019, a dengue outbreak occurred in our study area and was accompanied by an increase in the number of SDD and hospitalizations [35].

In the current study we investigated the influence of previous infection to ZIKV on acute clinical, virological, and immunological parameters in response to a subsequent DENV infection in an area where both viruses co-circulate. To overcome the barrier of cross-reactivity between flaviviruses and its impact in seroprevalence rates, we developed and validated a new peptide-based Enzyme-linked Immunosorbent Assay (ELISA) method for dengue and Zika.

## Material and methods

### Ethics statement

The study was conducted according to the guidelines of the Declaration of Helsinki and done in retrospective samples with the consent term waived approved by the Institutional Review Board of the School of Medicine of São José do Rio Preto (FAMERP) (protocol codes 15461513.5.0000.5415, approved on April 7, 2015 and 14262619.0.0000.5415, approved on August 13, 2019). Confidentiality was ensured by anonymizing all samples before data entry and analysis.

This cross-sectional study was based on data obtained from computerized medical records and disease notification forms for patients with suspected dengue enrolled in a health care unit-based acute febrile cohort between December 2018 and November 2019, São José do Rio Preto, Brazil. Collected data also included laboratory, molecular, virological, and immunological outcomes on collected blood samples.

### Study area

São José do Rio Preto, state of São Paulo, is an area where dengue is hyperendemic, with geo-climatic characteristics that favor its year-round circulation [as reviewed in [36]]. Other

arboviruses have been documented to circulate in the city, such as Saint Louis Encephalitis Virus (SLEV) [37], Chikungunya (CHIKV) [38], Ilheus (ILHV) [39] and ZIKV [40], as well as co-infections [41, 42]. São José do Rio Preto is among the top five cities in the state for reported dengue cases. The largest epidemics reported in the city occurred in 2010, 2013, 2015, 2016, and 2019 [35]. In 2016 Zika was introduced in the city causing an epidemic with 1,213 reported cases of which 267 were laboratory confirmed [43]. In 2019, with DENV-2 recirculating in the country, record numbers of dengue cases were recorded in the city: 33,120 cases, 528 with warning signs, 29 cases of severe dengue, and 19 deaths [35].

## Case eligibility for inclusion

This study was conducted using samples from patients with suspected dengue, classified according to WHO 2009 guidelines [5] and the Brazilian Ministry of Health [44]. After initial care was provided, the suspected cases were reported in SINAN (Brazilian notification system). Collected blood samples were subjected to relevant diagnostic assays (as indicated by the medical team) and stored at -80˚C until they were repurposed for additional testing relevant to this study. Selection for further testing also included samples from suspected DENV cases up to 7 days from the onset of symptoms. These samples were then tested using DENV RT-PCR. Detection of anti-dengue IgM antibodies (ELISA or immunochromatographic testing) was not considered diagnostic confirmation, given the existing possibility of cross-reaction with other flaviviruses [45, 46]. Samples that tested positive remained in the eligible pool, whereas negative samples were evaluated to detect the NS1 antigen. Samples from suspected DEN cases with negative DENV RT-PCR and negative NS1 antigen assay were excluded from subsequent analyses. In all patients with clinical information that could be obtained from the medical records or SINAN reporting forms, dengue was classified as: (i) dengue without warning signs (DwWS), (ii) dengue with warning signs (DWS), or (iii) severe dengue disease (SDD). Samples from these individuals were then evaluated for previous infection to dengue and Zika by testing for IgG antibodies to each flavivirus (**S1 Fig**).

**Definitions.** The definitions employed in this study are in accordance with national and international guidelines for clinical diagnosis and management of dengue [5, 44]. Patients were suspected to have dengue if they lived in or traveled in endemic areas and had fever and two or more of the following signs and symptoms: nausea, vomiting, exanthema, myalgia, arthralgia, positive tourniquet test, leukopenia, and some warning signs. Cases were considered confirmed when either NS1, or DENV RT-PCR tests were positive. Dengue with warning sings (DWS) was defined as the presence of at least one of the following signs or symptoms: abdominal pain, persistent vomiting, fluid accumulation, mucosal bleeding, lethargy, hepatomegaly greater than 2 cm, and elevated hematocrit with abrupt decreased platelet count. Finally, severe dengue disease (SDD) was classified as a suspected or confirmed case with severe plasma loss (characterized by shock and/or accumulation of fluids with respiratory distress), severe bleeding, severe organ dysfunction (characterized by liver enzymes alanine aminotransferase [ALT] and/or aspartate aminotransferase [AST] greater than 1000 IU/dl, impaired consciousness, and/or dysfunction of the myocardia or other organs). Previous dengue infection was defined by positive IgG anti-dengue (ELISA), and previous Zika infection by the presence of anti-Zika IgG (ELISA).

## Molecular methods for dengue investigation

To investigate acute dengue infection, viral RNA was extracted from each clinical sample using the QIAamp Viral RNAMini kit (Qiagen), following the manufacturer's instructions. The viral RNA was analyzed for the presence of the DENV genomic material, according to the

protocol described by Johnson et al. (2005) [47]. The DENV multiplex one-step reaction was performed with 10 μl of vRNA, 0.12 μl of each primer (DENV-1 and 3 to 50 μM and DENV-2 and 4 to 25 μM), 0.2 μl of each probe at 9 μM, 0.3 of Superscript III RT/Platinum Taq enzyme mixture (SuperScript III Platinum One-Step qRT-PCR System) and 6.25 of 2X PCR Master Mix applied to a plate of 96 wells using the QuantStudio Dx under the following conditions: 45˚C for 15 seconds, followed by 40 cycles of 95˚C for 15 seconds and 60˚C for 1 minute. The concentration of viral RNA was estimated in duplicate samples by quantification of complementary DNA using the single step qRT-PCR assay [46]. DENV control samples used to construct the standard curve were previously cultivated in Vero cells and titrated by plaque assay, and four points were established on the standard curve starting from the initial concentration from 103 to 100. The results were interpreted as positive when Ct values were less than or equal to 37.

To generate a viral quantification curve (equivalent viral titer), 10-fold dilutions (range 106–101 copies/mL) of DENV were prepared and tittered on Vero cells. Viral RNA was also extracted (from 1 ml of each of the dilutions prepared) using TRIzol RNA Isolation Reagents (Invitrogen), according to the manufacturer's recommendations, and used as a reference for the standard curve. The relative level of DENV RNA was determined by the $\Delta$Ct method, where Ct values for each sample were subtracted from Ct values from negative controls (2 $-\Delta$Ct) and expressed as relative abundance of viral (+)—stranded RNA. The viral plaques were counted to determine virus titer. Results were expressed as PFUeq/ml.

## Investigation of previous dengue or Zika infection

Samples from patients with confirmed dengue infection were evaluated for past history of dengue and Zika infection using an enzyme-linked immunosorbent assay (ELISA) developed in-house in partnership with the Gehrke's Lab at the Massachusetts Institute of Technology. The assay was specifically developed to eliminate the high cross-reactivity between flaviviruses observed in the commercial kits available on the market. For this, DENV1-4 and ZIKV NS1 libraries of 13-to 19-mers peptides, with variable amino acid overlaps, were tested against DENV and ZIKV specific monoclonal antibodies pairs described elsewhere [48]. After selection, the Peptide ELISA IgG assay (PepELISA) was analyzed with samples serologically validated by plaque reduction neutralization test (PRNT).

**Epitope libraries.** NS1 epitope screening was performed using peptide arrays from the Biodefense & Emerging Infectious (BEI) research resources repository provided by the National Institute of Allergy and Infectious Diseases (NIAID). DENV-1 61-peptide array spans the NS1 protein of strain Singapore/S275/1990 (GenPept: P33478), where the peptides are 13- to 17-mers, with 11 or 12 amino acid overlaps (Ref: NR-2751). DENV-2 47-peptide array spans the NS1 protein of strain New Guinea C (GenPept: AAA42941), where the peptides are 15-to 19-mers, with10 or 11 amino acid overlaps (Ref: NR-508). DENV-3 60-peptide array spans the NS1protein of strain Philippines/H87/1956 (GenPept: AAA99437), where peptides are 13-to 17-mers, with 11 or 12 amino acid overlaps (Ref: NR-2753). DENV-4 61-peptide array spans the NS1 protein of strain Singapore/8976/1995 (GenPept: AAV31422) where peptides are 13-to 17-mers, with 11or 12 amino acid overlaps (Ref: NR-2755). ZIKV 114-peptide array spans the nonstructural protein 1(NS1) of strain PRVABC59 (GenPept: AMZ03556), where peptides are 13-or15-mers, with 12 amino acid overlaps (Ref: NR-50534).

**Antibody production and purification.** Monoclonal antibodies were obtained as described [48]. Briefly, hybridoma cells producing antibodies against DENV and ZIKV NS1 were obtained by injection of C57BL/6 mice with DENV-1, DENV-2, DENV-3, DENV-4, and ZIKV. After hybridomas were screened using ELISA and fluorescent-activated cell sorting

(FACS), selected cell cultures were harvested and concentrated using Millipore centrifugal units (30 kDa MW). Protein L columns were used to purify the kappa light chain mouse antibodies that were specific to ZIKV and DENV NS1. After purification, the antibodies were buffer exchanged into PBS, concentrated, and stored at 4˚C. A NanoDrop 2000 UV−vis spectrophotometer at 280 nm was used to calculate the concentration of the purified antibody. A TapeStation with a P200 ScreenTape from Agilent Technologies was used to confirm the purity of the monoclonal antibodies.

**Validated sera cohort.** DENV-infected sera samples are originated from Southeastern Brazil outbreaks from 2006 to 2012, a period that precedes ZIKV introduction in this geographic area. ZIKV seropositive human samples were obtained from Fundação Ezequiel Dias (FUNED/MG) from archived serum samples collected between 2016 and 2017 representing RT-PCR-confirmed infections of patients with clinical follow-up. In total, 170 samples were previously tested by molecular and serological assays by the origin institution and retested for this work. For the DENV and ZIKV panels, samples were serologically tested by commercial ELISA kits, and PRNT was employed as a confirmatory post-test. Ethical Committees from the Faculdade de Medicina de São José do Rio Preto, SP (FAMERP), Fundação Ezequiel Dias (FUNED), and Universidade Federal de Minas Gerais (UFMG), approved the use of human sera in accordance with national regulations.

**Open sera cohort.** The sera samples that composed the hospital cohort were obtained between December 2018 and November 2019 at the Hospital de Base de São José do Rio Preto, SP, Brazil. The hospital is a reference center for infectious diseases for a region covering about 2,000,000 people. Dengue-confirmed patients by RT-PCR DENV or detection of NS-1 DENV antigen, with or without severity signs, according to Brazilian Ministry of Health's criteria [44], were eligible for screening.

**ELISA IgG.** DENV IgG was detected using the Panbio Dengue IgG Indirect ELISA (Alere, Waltham, MA, USA), according to the manufacturer's instructions. The assay did not discriminate between DENV serotypes. ZIKV IgG was detected using the anti-Zika Virus ELISA IgM/IgG assay (Euroimmun, Luebeck, Germany). The plate was read at 450 nm using a Spectramax Plus ELISA reader (Molecular Devices, LLC).

**PRNT.** Sera were serially 2-fold diluted, beginning with a 1/10 up to 1/160, and added to culture media solutions containing DENV or ZIKV reference strains. Each dilution was tested in duplicate, and the number of plaque-forming units (PFU) was recorded as the average of the number observed in each test. The PRNT50 titer is the highest serum dilution able to neutralize at least 50% of plaque formation when compared to infected cells in the absence of virus-positive serum. After PRNT validation, samples were segregated into DENV-1, DENV-2, DENV-3, DENV-4, and ZIKV positives. Samples from co-infected patients were excluded from the sera bank.

**NS1 peptide-ELISA (pepELISA).** The Peptide ELISA (PepELISA) protocol was developed *in house* and described elsewhere [49]. Briefly, we used flexible vinyl 96-well plates that were coated with 2ug/well of peptides and allowed to dry. Upon use, wells were blocked with bovine serum albumin (BSA) buffer. Sera samples were diluted in a 1:100 ratio in blocking buffer (BF) and incubated for 2 hours at 37˚C. After that, anti-human IgG antibody conjugated to peroxidase (Sigma, USA) was diluted in BF at a 1:10,000 ratio and incubated for 1 hour at 37˚C. Plates were developed with TMB (3,3',5,5' tetramethylbenzidine, Sigma, USA), and reactions were stopped with 0.5M sulfuric acid solution. The plate was read at 450 nm using a Spectramax Plus ELISA reader (Molecular Devices, LLC).

**NS1 ELISA.** In order to compare the diagnostic performance of the peptides, an ELISA with NS1 protein for both viruses were performed. For this, an *in house* assay was established following modifications of an existing protocol [50]. Briefly, 96-well plates (Thermo Fisher Scientific Inc, USA) were individually coated with 200 ng per well of recombinant NS1 DENV

and NS1 ZIKV proteins (The Native Antigen Company, UK). Diluted patient sera (1:50) were added to the wells and incubated at RT for 60 min. Plates were, then, incubated with anti-human IgG antibody conjugated to peroxidase (Sigma, USA), diluted 1:5,000 at RT for 60 min, and reactions terminated with 0.5M sulfuric acid solution. The plate was read at 450 nm using a Spectramax Plus ELISA reader (Molecular Devices, LLC).

**Statistical analysis.** All analyses were performed in GraphPad Prism, version 9. To evaluate the potential of the selected peptides for DENV/ZIKV diagnosis, we calculated the sensitivity, specificity, accuracy, area under ROC curve, and the positive and negative predictive values [50]. To evaluate the performance of the ELISA, Receiver Operating Characteristic (ROC) curves were created using GraphPad Prism 9.0 software. The ROC curve presents test performance as a True Positive Rate (%sensitivity) versus a False Positive Rate (100%—%specificity). Optimal cutoff values, which maximize sensitivity and specificity, were calculated from the ROC curve using GraphPad Prism 9.0. Sensitivity is defined as the fraction of total confirmed positive samples that are true positives according to the test. Specificity is defined as the fraction of total confirmed negative samples that are true negatives according to the test. Confidence intervals (CI) using the Wilson/Brown method and Area Under Curve (AUC) were calculated for each analysis using GraphPad. The Kolmogorov-Smirnov method was used as a normality test to evaluate data distribution. The ANOVA test with Bonferroni correction as a multiple hypothesis test was used to compare groups with parametric distribution, and Kruskal-Wallis test with Dunn's correction was used to compare data with non-parametric distribution.

For the following statistical analyses four groups of samples were defined: i) DV (dengue)-/ZV(Zika)- (control); ii) DV-/ZV+; iii) DV+/ZV-; iv) DV+/ZV+. Since the objective of this study was to evaluate the influence of anti-Zika IgG antibodies on the clinical course of acute dengue, the DV+/ZV+ group was excluded from subsequent analyses, because it was not possible to determine whether the effect on disease progression was the influence of anti-dengue or anti-Zika IgG antibodies.

## Evaluation of cytokine expression profile

A commercial panel was used to determine the levels of circulating cytokines in the serum from the patient samples. For this purpose, a multiplex system analysis was performed using the Human Cytokine/Chemokine Magnetic Bead Panel commercial kit (Millipore, Watford, UK). The levels of thirteen molecules were measured according to the manufacturer's instructions: epidermal growth factor (EGF), tumoral necrosis factor (TNF)-α, interferon (IFN)-γ, interleukin (IL) -1ra, IL-1β, IL-2, IL-4, IL-6, IL-7, IL-8, IL-10, IL-13, and IL-17A. The beads were washed and analyzed using the Luminex IS-100 system (Luminex Corp., Texas, USA). The standard curves of known concentrations of recombinant human cytokines were used to convert fluorescence units into cytokine concentration units (pg/mL). Data were stored and analyzed using GraphPad Prism software, version 9.0 (GraphPad Software, San Diego, CA, USA), and subsequent correlated with previous Zika or dengue infection.

## Statistical analysis

To influence of anti-Zika antibodies in dengue infection, the data were analyzed for normality distribution using the Shapiro-Wilk test. ANOVA with the Bonferroni post-hoc test was used to evaluate the significance of differences between each group in normally distributed data. When normality was not met, the Kruskal-Wallis test was used to determine whether there was a difference between the means of the groups. Pearson's chi-square test was used to determine whether the expected frequency in the groups was met. Finally, binary logistic regression was performed to verify the predictors of hospitalization or severe forms (dengue with warning

signal or severe dengue) between the control groups and the groups with a history of dengue or Zika infection. To select the variables that would comprise the final model, a discriminant analysis was performed with p<0.1, estimated by Rao's score test. The variables that obeyed the predefined criteria were subjected to multivariate analysis, with significance defined as p<0.05. A 0.05 (5%) significance level was adopted for all tests, and all data were tabulated and analyzed with SPSS software for IOS (version 28, SPSS, Inc; Chicago, Il, USA).

## Results

### Development and validation of PepELISA

Five epitope libraries, totaling 343 NS1 peptides, were screened using detection antibodies previously described as serotype-specific for NS1 ELISA of Dengue virus [48], with the addition of ZIKV-specific antibodies (**Fig 1**). A first screening was performed using the provided BEI library, in a 96-well array, where highest scores are shown in **Fig 1A**. Peptides originated from the ZIKV library were designated as ZV plus the position number at the original library. Similar designations were used for the DENV library, with the addition of a serotype identifier (DV-1, DV2, DV3, or DV4). We selected the peptides ZV-53, ZV-54, ZV-107, DV2-15, and DV4-20 for soluble synthesis, integrity analysis, and validation of the pepELISA platform.

The immobilization of small proteins or peptides on surfaces has been a barrier to the development of reproductible and reliable tests. The methodology of drying the peptides in a vinyl surface has been used as a method of mimicking an ELISA assay, called here as pepELISA, and first described elsewhere [49]. Following soluble synthesis, only peptide ZV-53 was unstable and considered unsuitable for the consequent analysis. As observed at **Fig 2B**, DENV peptides were recognizable only by DENV-specific antibodies (in green), with low to absent signal against ZIKV-specific antibodies (in blue). As expected, ZIKV peptides behave in the exact opposite, being recognizable only by ZIKV-specific antibodies. Is important to note that ZV-107 reacts strongly to mAb 130 (see **Fig 2B**), but not to other ZIKV antibodies.

After initial testing, peptides were further evaluated using human sera. To this end, we used 170 samples, sourced from DENV seropositive patients, ZIKV seropositive patients, and sera from flavivirus seronegative subjects. These samples where previously screened by neutralization assays, considered as the gold standard for flaviviruses serology and serotyping. We utilized this sera cohort not only to test the diagnostic performance of the pepELISAs, but also to evaluate the same performance in a standard protein setting (called here NS1 ELISA) (**Fig 2**). Peptide ZV-107 was excluded from the analysis after not differentiating seronegative from positive. As expected, both synthetic peptides were able to efficiently separate DENV positive from ZIKV positive patients (**Fig 2A and 2C**). However, although recognizing flavivirus positive sera in comparison to seronegative subjects, the DENV NS1 ELISA setting was not able to successfully identify only DENV-specific sera. The ZIKV NS1 ELISA, on the other hand, had the worst diagnostic performance among all assays, with close to 50% of positive samples being identified as negative.

The cutoff value of pepELISA was obtained by the ROC curve and determined by the highest far left point of the curve, which gives the highest sensitivity and specificity values. From the ROC curve it is also possible to analyze the Area under the Curve (AUC), a measure that represents the accuracy or overall performance of the test, as it considers all sensitivity and specificity values for each cutoff point. Overall, the closer AUC is to 1, the better the diagnostic performance of the assay. Performance analysis for each antigen is presented at **Table 1.**

Finally, both peptides presented higher sensitivity compared to the standard ELISA performed with the whole protein (NS1). Peptide DV-15 obtained 83% of sensitivity and 95% specificity. Peptide DV-20, however, showed a higher sensitivity rate of 96%. The ZIKV

**(A)**

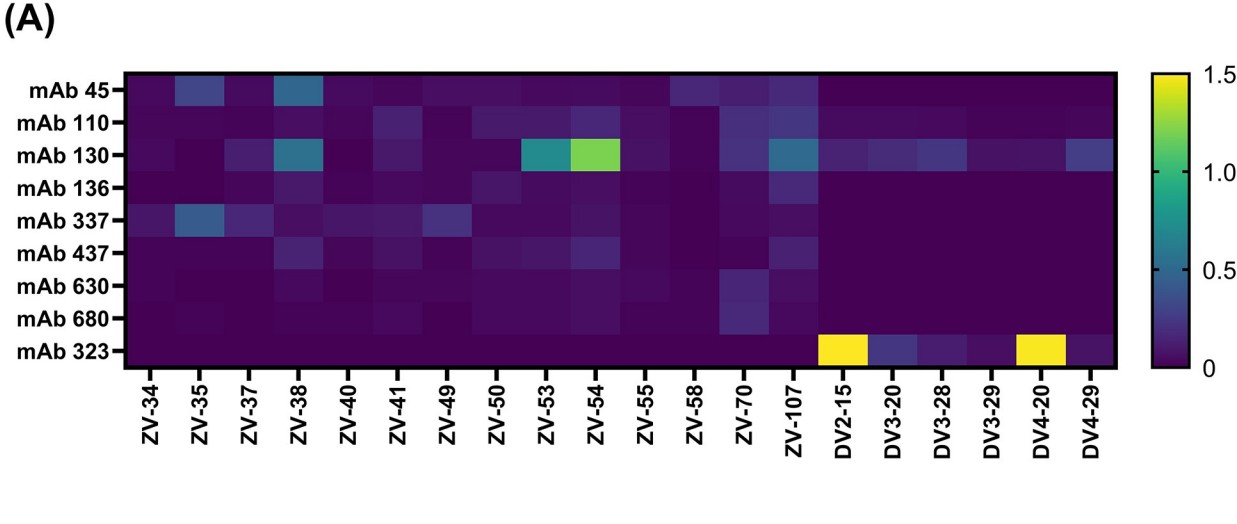

**(B)**

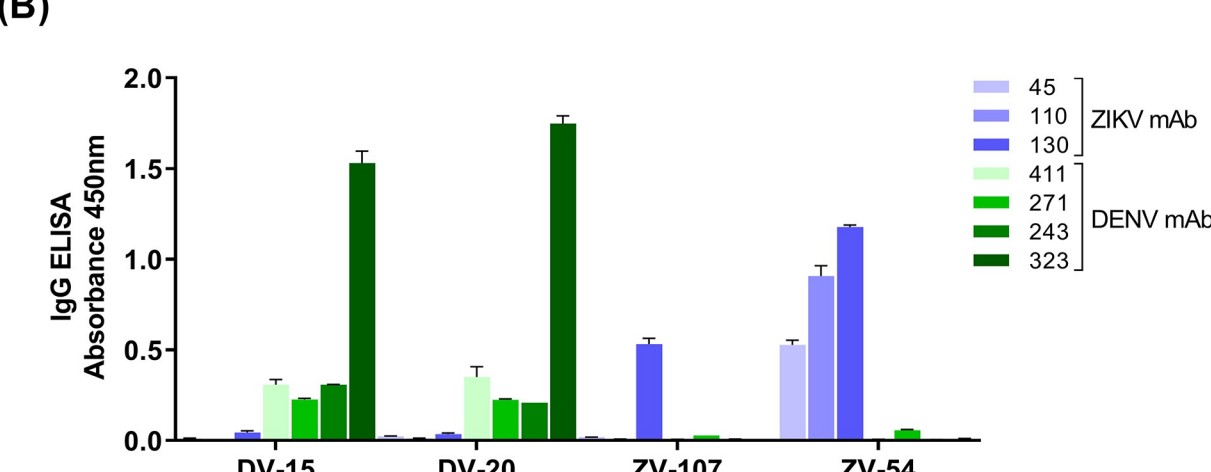

**Fig 1. Screening of reactive peptides using monoclonal antibodies. (A)** Heatmap matrix of peptides assessed with monoclonal specific antibodies against ZIKV or DENV in an indirect IgG ELISA platform. Highest scores were selected for soluble synthesis. Peptide ZV-53 was not stable in soluble form and excluded from analysis. **(B)** IgG pepELISA for validation of soluble peptides. Serum samples from patients validated with PRNT were evaluated on plates with DV-15, DV-20, ZV-54, and ZV-107.

peptide obtained the better overall performance, with 97% sensitivity and specificity (**Fig 3**). Although able to separate seronegative samples, both protein assays performed poorly when compared to the correlate peptide. Additionally, we analyzed the performance of the peptide-versus protein-based assays in differentiating DENV from ZIKV. We incorporated DENV-negative but ZIKV-positive sera in ROC analyses for the DV-20 peptide versus DENV-NS1 assays. Likewise, we incorporated ZIKV-negative but DENV-positive sera into ROC analyses for the ZV-54 peptide versus ZIKV-NS1 assays. In this differential analysis (**S2 Fig**), peptide-based assays for both pathogens yielded AUCs substantially higher than those for NS1 ELISAs (>0.94 versus 0.66–0.74) (**S2 Table**). Ultimately, DV-20 (96% sensitivity and 98% specificity) and ZV-54 (94% sensitivity and 88% specificity) peptides are improved antigenic targets to differentiate dengue-specific from Zika-specific antibodies.

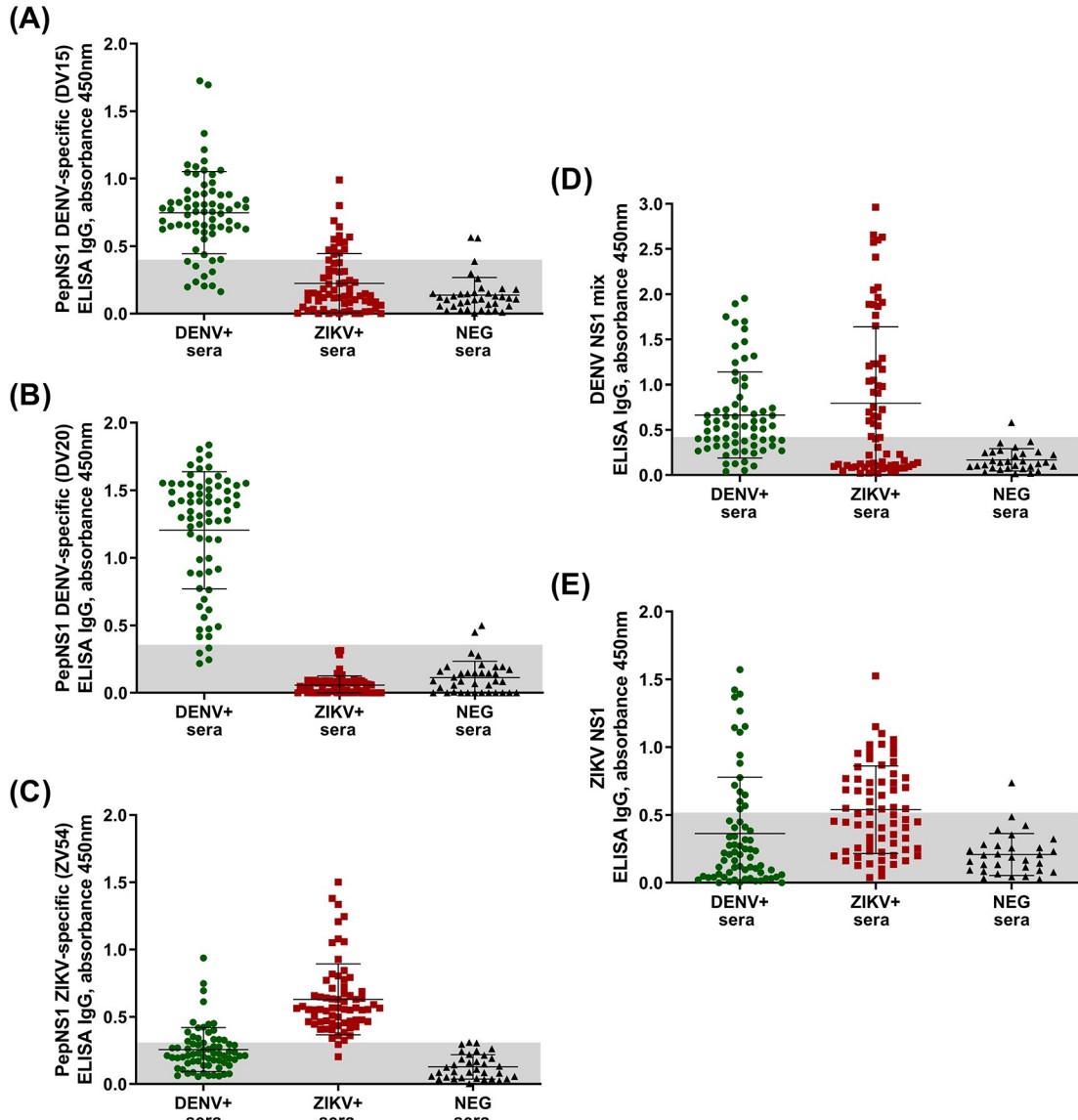

**Fig 2. PepELISA and standard ELISA using validated samples.** A total of 170 serum samples validated with PRNT were tested on plates with: **(A)** DV-15; **(B)** DV-20; and **(C)** ZV-54. Peptide ZV-107 did not recognize any of the serum samples and was excluded from the further analysis. The same panel of samples were also tested with: **(D)** a mix of the NS1 protein of DENV-1, DENV-2, DENV-3, and DENV-4; and **(E)** To complete the positive controls, we tested the samples in plates coated with ZIKV NS1 protein. Gray area indicates the cutoff values based on the ROC curve specific for each antigen (**Fig 3**). Diagnostic performance analysis and cutoff values are presented in **Table 1**. Samples within the gray area are considered negative.

## General characteristics of the study population

Between December 2018 and November 2019, 2,990 serum samples from patients with suspected dengue were obtained, of which 1,490 were laboratory confirmed using DENV-specific RT-PCR or dengue NS1 antigen assays. DENV-2 was the most frequent serotype identified (99.8%). Of these, 1,043 were selected based on the availability of medical records to classify the severity of the clinical presentation, and past history of dengue and Zika infection by ELISA. Most of the cases occurred during the first quarter of 2019, in line with the expected seasonality, although a few cases were documented in the remaining months (**S3 Fig**) Samples

**Table 1. Measure of diagnostic performance for DENV NS1 protein, ZIKV NS1 protein, and for peptides DV-15, DV-20, and ZV-54.**

|                  | NS1 DENV        | NS1 ZIKV        | DV-15            | DV-20            | ZV-54            |
|------------------|-----------------|-----------------|------------------|------------------|------------------|
| AUC              | 0.9044          | 0.823           | 0.9803           | 0.9925           | 0.9953           |
| 95% CI           | 0.8443 to 0.9645 | 0.7413 to 0.9048 | 0.9604 to 1.000 | 0.9823 to 1.000  | 0.9872 to 1.000  |
| Cutoff           | > 0.42          | > 0.52          | > 0.46           | > 0.31           | > 0.31           |
| Sensitivity (%)  | 60.29           | 47.83           | 82.86            | 95.71            | 97.1             |
| Specificity (%)  | 96.77           | 96.77           | 94.59            | 94.59            | 97.06            |
| N Total Positive | 31              | 31              | 70               | 70               | 69               |
| N Total Negative | 68              | 69              | 37               | 37               | 34               |

were classified into four groups: (1) naïve to DENV and ZIKV (DV-/ZV-; control), n = 203 (19.46%); (2) DV-/ZV+, n = 49 (4.69%); (3) DV+/ZV-, n = 627 (60.11%); and (4) DV+/ZV+, n = 164 (15.72%) (**Fig 4**).

The demographic and clinical characteristics of the 1,043 cases are presented in **S1 Table.** Mean patient age was 40.75 years (±18.74), and they were predominantly female (54.3%). The

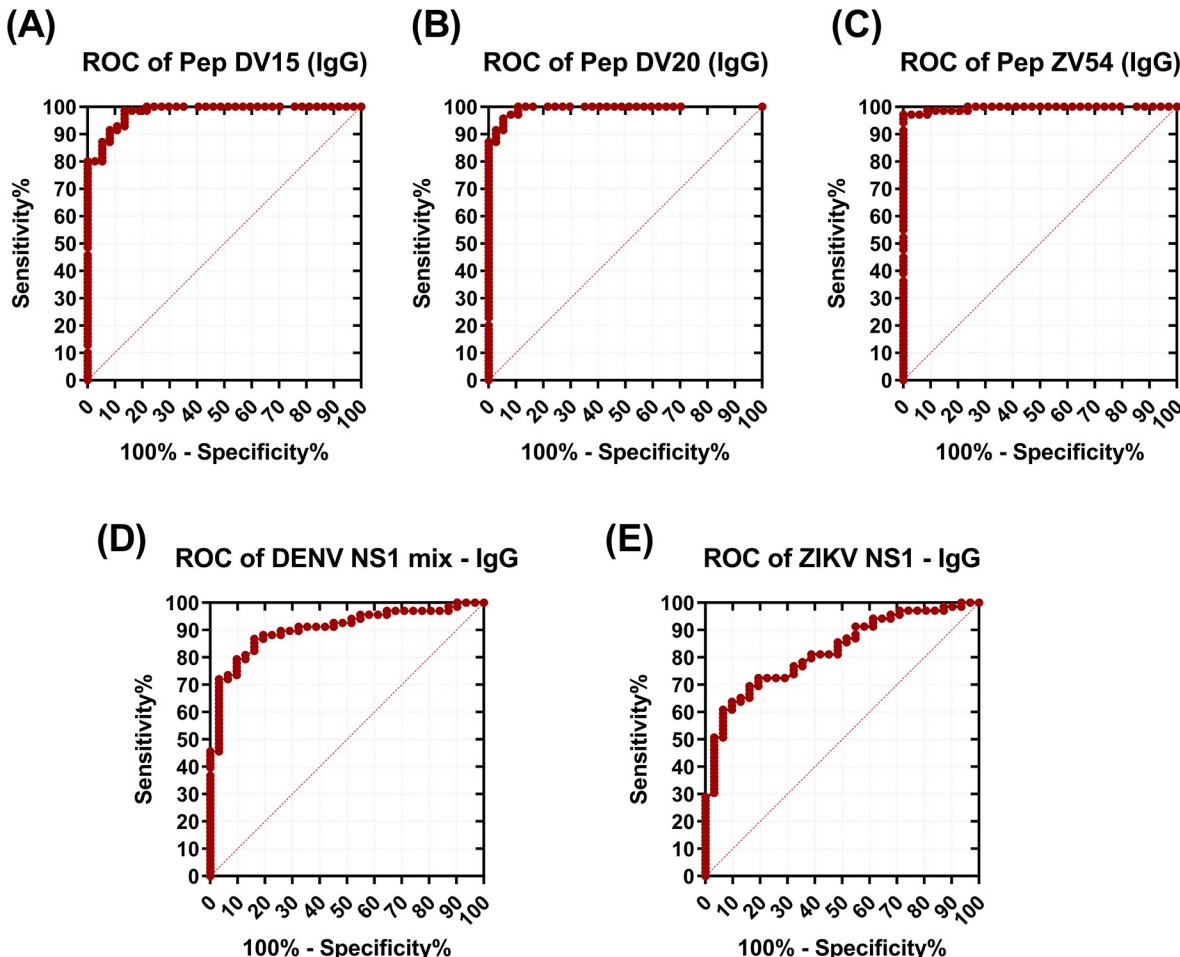

**Fig 3. Performance of the pepELISA and standard NS1 ELISA using validated samples.** Receiver Operating Characteristic (ROC) curves of pepELISA for the peptides: (**A**) DV-15; (**B**) DV-20; (**C**) ZV-54; and standard ELISA against DENV NS1 mix (**D**), and ZIKV NS1 (**E**). Performance is demonstrated in True Positive Rate (Sensitivity %) versus False Positive Rate (100%—Specificity %).

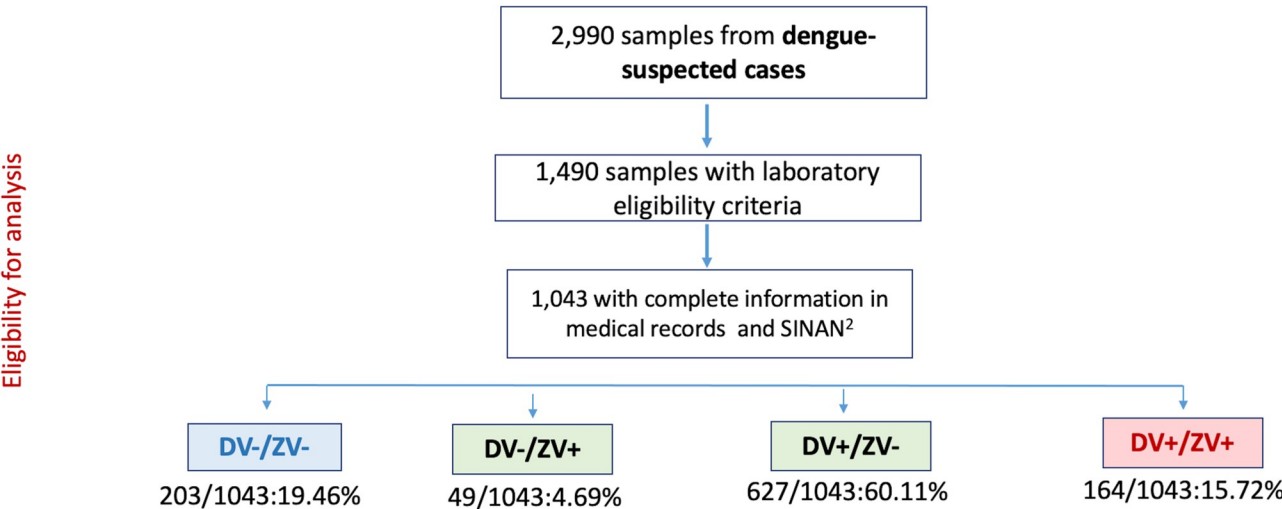

**Fig 4. Samples selection and case distribution according to serological sub cohort.** Criteria for inclusion into the cohort included laboratory-confirmed infection by RT-PCR and/or NS-1 ELISA or immunochromatographic method.

most frequently reported symptoms were fever (84.0%), myalgia (82.6%), and headache (74.2%) and the most frequent warning signs were abdominal pain (40.2%), followed by hemo-concentration (39.9%) and non-severe bleeding (23.1%). Central nervous system involvement (1.6%), signs of shock (1.6%), and severe bleeding (1.2%) were the most prevalent signs of severity. Based on the 2009 WHO Dengue Clinical Classification (5), 820 cases were classified as dengue without warning signs (DwWS) (78.6%), 212 dengue with warning signs (DWS) (20.3%) and 11 cases as SDD (1.1%). A total of 30.2% of patients required hospitalization with a mortality rate of 0.2 deaths per 100 cases. Because of the difficulty in determining the specificity of the previous infection in the group with Zika and dengue infections (DV+/ZV + group), these samples were excluded from further analysis.

## Relationship between previous dengue or Zika and impact on acute dengue

The groups of interest (DV-/ZV-, DV-/ZV+, and DV+/ZV-) comprised a total of 879 cases; the demographic and clinical characteristics and relevant analysis are presented in Tables 1 and 2.

No difference was observed between gender (p = 0.100) or mean age (p = 0.365) between the patients that formed the three groups. Leukopenia was more frequent in the DV-/ZV + group (p<0.001), although no difference was observed in mean leukocyte count; no statistically significant differences were observed for hematocrit, transaminases, alkaline phosphatase, creatinine, or albumin. Additionally, lower mean platelet count was observed in the DV-/ZV + group (p = 0.013), although the frequency of thrombocytopenia did not vary among the groups (**Table 2**).

Although no differences were observed in the overall frequencies of symptoms suggestive of dengue with or without warning signs among the groups, severe bleeding was more frequent in the DV-/ZV+ patients (p = 0.036; 2/31 (6.45%) in DV-/ZV+ vs. 0% in control). When the cases were clinically classified, there was a higher occurrence of severe and potentially severe forms (SDD+DWS) in the DV-/ZV+ group (p<0.001). The hospitalization rate was 68% in the DV-/ZV+ group, versus 39% in DV-/ZV- and 24% in DV+/ZV- (p<0.001). Although hospitalization time did not differ among the groups, the time between appearance of symptoms and

**Table 2. Demographic characteristics, frequency of signs and symptoms compatible with dengue and laboratory findings of 879 patients of sub cohorts.**

| | DV-ZV- | | | DV-/ZV+ | | | DV+/ZV- | | |
|---|---|---|---|---|---|---|---|---|---|
| | N samples | N positive or mean | % or s.d. | N samples | N positive or mean | % or s.d. | N samples | N positive or mean | % or s.d. | p-value |
| Age mean | 201 | 38.2 | 19.9 | 48 | 41.06 | 18.18 | 615 | 40.17 | 18.16 | 0.365 |
| **Age group** | | | | | | | | | | |
| 15–59 years of age | 201 | 141 | 70.1% | 48 | 36 | 75.0% | 615 | 472 | 76.7% | 0.121 |
| < 15 years of age | 201 | 27 | 13.4% | 48 | 3 | 6.25% | 615 | 47 | 7.7% | |
| >60 years of age | 201 | 33 | 16.4% | 48 | 9 | 18.75% | 615 | 96 | 15.6% | |
| **Gender** | | | | | | | | | | |
| Female | 203 | 101 | 49.8% | 49 | 22 | 44.9% | 627 | 354 | 56.5% | 0.100 |
| Male | 203 | 102 | 50.2% | 49 | 27 | 55.1% | 627 | 273 | 43.5% | |
| **Signs and symptoms** | | | | | | | | | | |
| Fever | 199 | 164 | 82.41% | 46 | 44 | 95.65% | 619 | 513 | 82.87% | 0.072 |
| Myalgia | 198 | 151 | 76.26% | 46 | 37 | 80.43% | 618 | 513 | 83.00% | 0.104 |
| Arthralgia | 199 | 38 | 19.09% | 46 | 8 | 17.39% | 619 | 128 | 20,67% | 0.793 |
| Nausea | 199 | 64 | 32.16% | 46 | 18 | 39.13% | 619 | 241 | 38.93% | 0.222 |
| Headache | 199 | 134 | 67.33% | 46 | 35 | 76.08% | 619 | 466 | 75.28% | 0.080 |
| Ocular pain | 199 | 98 | 49.24% | 46 | 21 | 45.65% | 619 | 318 | 51.37% | 0.689 |
| Vomit | 199 | 41 | 20.60% | 46 | 8 | 17.39% | 619 | 109 | 17.60% | 0.628 |
| Exanthema | 198 | 61 | 30.80% | 46 | 11 | 23.91% | 619 | 178 | 28.75% | 0.772 |
| Leukopenia (<5,000 cell/mm$^3$) | 203 | 79 | 38.91% | 49 | 27 | 55.10% | 624 | 135 | 21.63% | **<0.001** |
| Thrombocytopenia (<150x10$^3$/mm$^3$) | 97 | 58 | 59.79% | 33 | 26 | 78.79% | 149 | 85 | 57.04% | 0.068 |
| Tourniquet test positive | 158 | 27 | 17.08% | 36 | 2 | 5.55% | 552 | 102 | 18.47% | 0.140 |
| **Laboratory tests** | | | | | | | | | | |
| Ht mean (%) | 98 | 40.93 | 4.74 | 35 | 40.79 | 7.32 | 150 | 40.47 | 4.56 | 0.771 |
| Hb mean (g/dl) | 98 | 13.95 | 1.70 | 35 | 14.04 | 2.63 | 150 | 13.98 | 1.60 | 0.966 |
| Leukocyte mean (cells/mm$^3$) | 98 | 5,454 | 2,904 | 35 | 4,858 | 2,109 | 150 | 5,456 | 5,129 | 0.731 |
| Platelet mean (plat/mm$^3$) | 98 | 118,571 | 89,548 | 35 | 88,91 | 88,719 | 150 | 135,327 | 82,614 | **0.013** |
| Albumin mean (g/dl) | 51 | 3.72 | 0.49 | 25 | 3.59 | 0.555 | 61 | 3,73 | 0.60 | 0.418 |
| Creatinine mean (mg/dl) | 32 | 1.23 | 0.74 | 10 | 0.85 | 0.22 | 62 | 0.96 | 0.46 | 0.074 |
| AST mean (UI/l) | 63 | 212.44 | 710.19 | 26 | 128.38 | 110.78 | 79 | 271.75 | 1.429,90 | 0.767 |
| ALT mean (UI/l) | 67 | 82.63 | 173.51 | 26 | 78.26 | 72.99 | 88 | 138.43 | 551.92 | 0.918 |
| AP mean (UI/l) | 33 | 103.36 | 61.01 | 16 | 106.13 | 93.63 | 36 | 114.94 | 91.20 | 0.658 |
| Ct mean | 127 | 27.03 | 5.21 | 27 | 27.90 | 5.88 | 0.372 | 325 | 24.84 | 5.33 |
| Viral titer (PFUeq/ml) | 126 | 577.38 | 2,515.33 | 27 | 187.28 | 426.98 | 0.693 | 325 | 761.24 | 2,262,73 |

hospital admission was shorter in the DV+/ZV- group (p = 0.029). Finally, no statistically significant differences were observed in the outcome of dengue infection (resolution vs. death) among the groups (**Table 3**). These results suggest that previous Zika infection was associated with severe disease outcomes following a DENV infection.

Logistic regression analyses showed that patients with a history of Zika infection had a higher risk (2.343 times higher, CI 95% 1.239–4429) of developing severe forms of dengue disease (DWS+SDD), and 3.390 times higher risk (CI 95% 1.594–7209) of hospitalization compared to the controls (DV-/ZV-) (**Table 4**). Furthermore, advanced age (>59 years) was a risk factor for severe forms (DWS+SDD) of dengue disease and hospitalization (OR 2.12; CI 95% 1.702–2.641; p<0.001; OR 2.202; CI 95% 1.729–2.806; p<0.001, respectively), where males were more likely to be hospitalized (OR 1.246; CI 95% 1.045–1.484; p = 0.014). The

**Table 3. Frequency of warning and severity signs of dengue and clinical outcome of 879 patients of sub cohorts.**

| | DV-ZV- | | | DV-/ZV+ | | | DV+/ZV- | | | |
|---|---|---|---|---|---|---|---|---|---|---|
| | N samples | N positive or mean | % or s.d | N samples | N positive or mean | % or s.d. | N samples | N positive or mean | % or s. d. | p-value |
| Time between symptom's onset and WS | 56 | 4.57 | 5.03 | 20 | 4 | 2.17 | 83 | 3.48 | 2.00 | 0.184 |
| Ct mean | 127 | 27.03 | 5.21 | 27 | 27.90 | 5.88 | 0.372 | 325 | 24.84 | 5.33 |
| Viral titer (PFUeq/ml) | 126 | 577.38 | 2,515.33 | 27 | 187.28 | 426.98 | 0.693 | 325 | 761.24 | 2,262,73 |
| **Warning signs** | | | | | | | | | | |
| Abdominal pain | 75 | 28 | 37.33% | 34 | 22 | 64.70% | 100 | 45 | 45% | 0.525 |
| Persistent emesis | 75 | 4 | 5.33% | 30 | 1 | 3.33% | 100 | 0 | 0.00% | 0.073 |
| Lethargy | 75 | 1 | 1.33% | 30 | 1 | 3.33% | 100 | 7 | 7% | 0.185 |
| Bleeding | 75 | 17 | 22.66% | 30 | 9 | 30.00% | 100 | 22 | 22% | 0.650 |
| Hepatomegaly | 75 | 6 | 8.00% | 30 | 1 | 3.33% | 100 | 6 | 6% | 0.662 |
| Hemoconcentration | 62 | 26 | 41.93% | 16 | 8 | 50.00% | 110 | 39 | 35.45% | 0.445 |
| Fluid accumulation | 75 | 10 | 13.33% | 30 | 6 | 20.00% | 100 | 15 | 15% | 0.689 |
| Hypotension | 36 | 0 | 0.00% | 11 | 1 | 9.09% | 58 | 5 | 8.6% | 0.190 |
| **Severity signs** | | | | | | | | | | |
| Severe bleeding | 75 | 0 | 0.00% | 31 | 2 | 6.45% | 100 | 1 | 1.00% | **0.036** |
| Respiratory distress | 75 | 0 | 0.00% | 31 | 1 | 3.23% | 100 | 1 | 1.00% | 0.305 |
| Shock | 75 | 0 | 0.00% | 31 | 1 | 3.23% | 100 | 1 | 1.00% | 0.305 |
| Vasoactive drug use | 75 | 0 | 0.00% | 31 | 0 | 0.00% | 100 | 1 | 1.00% | 0.587 |
| Liver disfunction | 75 | 0 | 0.00% | 31 | 0 | 0.00% | 100 | 0 | 0.00% | NA |
| Seizures | 75 | 0 | 0.00% | 31 | 0 | 0.00% | 100 | 2 | 2.00% | 0.343 |
| Encephalopathy | 75 | 0 | 0.00% | 31 | 0 | 0.00% | 100 | 1 | 1.00% | 0.587 |
| Guillain-Barré Syndrome | 75 | 0 | 0.00% | 28 | 1 | 3.57% | 45 | 1 | 2.22% | 0.408 |
| Laboratory manifestation* | 76 | 54 | 71.05% | 25 | 21 | 84.00% | 132 | 78 | 59.09% | 0.207 |
| Hemorrhagic manifestation | 75 | 17 | 22.66% | 31 | 10 | 32.25% | 100 | 23 | 23.00% | 0.530 |
| CNS involvement | 75 | 0 | 0.00% | 31 | 1 | 3.23% | 100 | 3 | 3.00% | 0.310 |
| **Current dengue infection** | | | | | | | | | | |
| DwWS | 203 | 144 | 70.9% | 49 | 25 | 51.0% | 627 | 523 | 83.4% | **<0,001** |
| DWS | 203 | 59 | 29.1% | 49 | 20 | 40.8% | 627 | 99 | 15,80% | |
| SDD | 203 | 0 | 0,00% | 49 | 4 | 8.2% | 627 | 5 | 0.8% | |
| DWS + SDD | 203 | 59 | 29.1% | 49 | 24 | 49% | 627 | 104 | 16.6% | **<0,001** |
| Hospitalization | 159 | 62 | 39% | 38 | 26 | 68.4% | 434 | 106 | 24.4% | **<0,001** |
| Time between symptoms onset and Hospitalization (in days) | 58 | 4,03 | 5,09 | 16 | 4,25 | 2,56 | 106 | 2,68 | 2,26 | **0,029** |
| Hospital staying average (in days) | 61 | 3,51 | 2,48 | 16 | 4,31 | 2,36 | 108 | 3,74 | 4.22 | 0,724 |
| DWS + SDD or hospitalization | 203 | 71 | 34.9% | 49 | 29 | 59.2% | 627 | 120 | 19.1% | **<0,001** |
| **Clinical outcome** | | | | | | | | | | |
| Resolution | 184 | 184 | 100,00% | 39 | 39 | 100,00% | 604 | 603 | 99,70% | 0,691 |
| Death | 184 | 0 | 0,00% | 39 | 0 | 0,00% | 604 | 2 | 0,30% | |

* Hemoconcentration, leukopenia or thrombocytopenia

multivariate model used to assess whether age, gender, and previous Zika infection were predictors for the development of severe forms of dengue disease showed significance for age >59 (OR 2.379; CI 95% 1.712–3.306) and previous Zika infection (OR 2.019; CI 95% 1.138–3.584), but not gender. Similar findings were observed for hospitalization, namely advanced age (OR 2.204; CI 95% 1.551–3.132; p<0.001) and previous Zika (OR 2.865; CI 95% 1.494–5.495; p = 0.002) associated with greater risk for hospitalization from acute dengue.

**Table 4. Risk to development of severe forms and hospitalization in dengue.**

| Risk of DWS+SDD | p-value | OR | CI 95% min | CI 95% max |
|---|---|---|---|---|
| *Univariate analysis* | | | | |
| DV-/ZV- | - | 1 | - | - |
| DV-/ZV+ | **0,009** | **2.343** | **1,239** | **4.429** |
| DV+/ZV- | **<0,001** | **0,485** | **0,336** | **0,702** |
| Age 15–59 years of age | - | 1 | - | - |
| Age < 15 years of age | **0.002** | **0.205** | **0.074** | **0.569** |
| Age > 59 years of age | **< 0.001** | **2.291** | **1.610** | **3.259** |
| Female | - | 1 | - | - |
| Male | 0.364 | 0.871 | 0.646 | 1.174 |
| *Multivariate analysis* | | | | |
| DV-/ZV+ | **0,004** | **2.149** | **1,107** | **4.175** |
| Age > 59 years of age | **< 0.001** | **2.486** | **1.657** | **3.730** |
| **Risk of hospitalization** | | | | |
| *Univariate analysis* | | | | |
| DV-/ZV- | - | 1 | - | - |
| DV-/ZV+ | **0,002** | **3.390** | **1.594** | **7.209** |
| DV+/ZV- | **0,001** | **0,506** | **0,343** | **0,744** |
| Age 15–59 years of age | - | 1 | - | - |
| Age < 15 years of age | **< 0.001** | **0.641** | **0.419** | **0.980** |
| Age > 59 years of age | **< 0.001** | **2.202** | **1.729** | **2.806** |
| Female | - | 1 | - | - |
| Male | 0.014 | 1.246 | 1.045 | 1.484 |
| **Dengue symptoms** | | | | |
| Fever | 0.086 | 1.511 | 0.943 | 2.420 |
| Myalgia | 0.073 | 0.686 | 0.454 | 1.036 |
| Arthralgia | 0.073 | 1.416 | 0.968 | 2.072 |
| Nausea | 0.134 | 0.772 | 0.550 | 1.083 |
| Headache | <0.001 | 0.278 | 0.195 | 0.396 |
| Ocular pain | <0.001 | 0.391 | 0.280 | 0.546 |
| Emesis | 0.083 | 1.430 | 0.955 | 2.142 |
| Exanthema | 0.014 | 0.621 | 0.424 | 0.909 |
| Tourniquet test positive | 0.012 | 0.376 | 0.176 | 0.804 |
| Thrombocytopenia | <0.001 | 9.341 | 5.250 | 16.622 |
| Leukopenia | <0.001 | 9.312 | 6.478 | 13.387 |
| *Multivariate analysis* | | | | |
| Thrombocytopenia | **< 0.001** | **8.442** | **4.584** | **15.549** |

To understand the potential mechanisms contributing to the clinical exacerbation of dengue from prior ZIKV infection, we evaluated various viral and immunological factors. Dengue viral titer was found to be similar in the DV-/ZV+ group compared to the control group (DV-/ZV-) (p = 0.372 for mean Ct and p = 0.693 for equivalent viral titer). Interestingly, despite no difference in clinical presentations, the viral load (both Ct and equivalent viral titer) was higher in the DV+/ZV- group than in the control group (**Table 3**).

To investigate the role of acute immune response in response to DENV infection, we measured concentrations of cytokines (interferon (IFN)-g, IL-1β, IL-2, IL-4, IL-6, IL-8, IL-9, IL-10, IL-13, and IL-17) and chemokines in plasma obtained at the time of admission (**Fig 5**). Previous dengue infection was associated with increased concentrations of IL-17A and IL-1β but

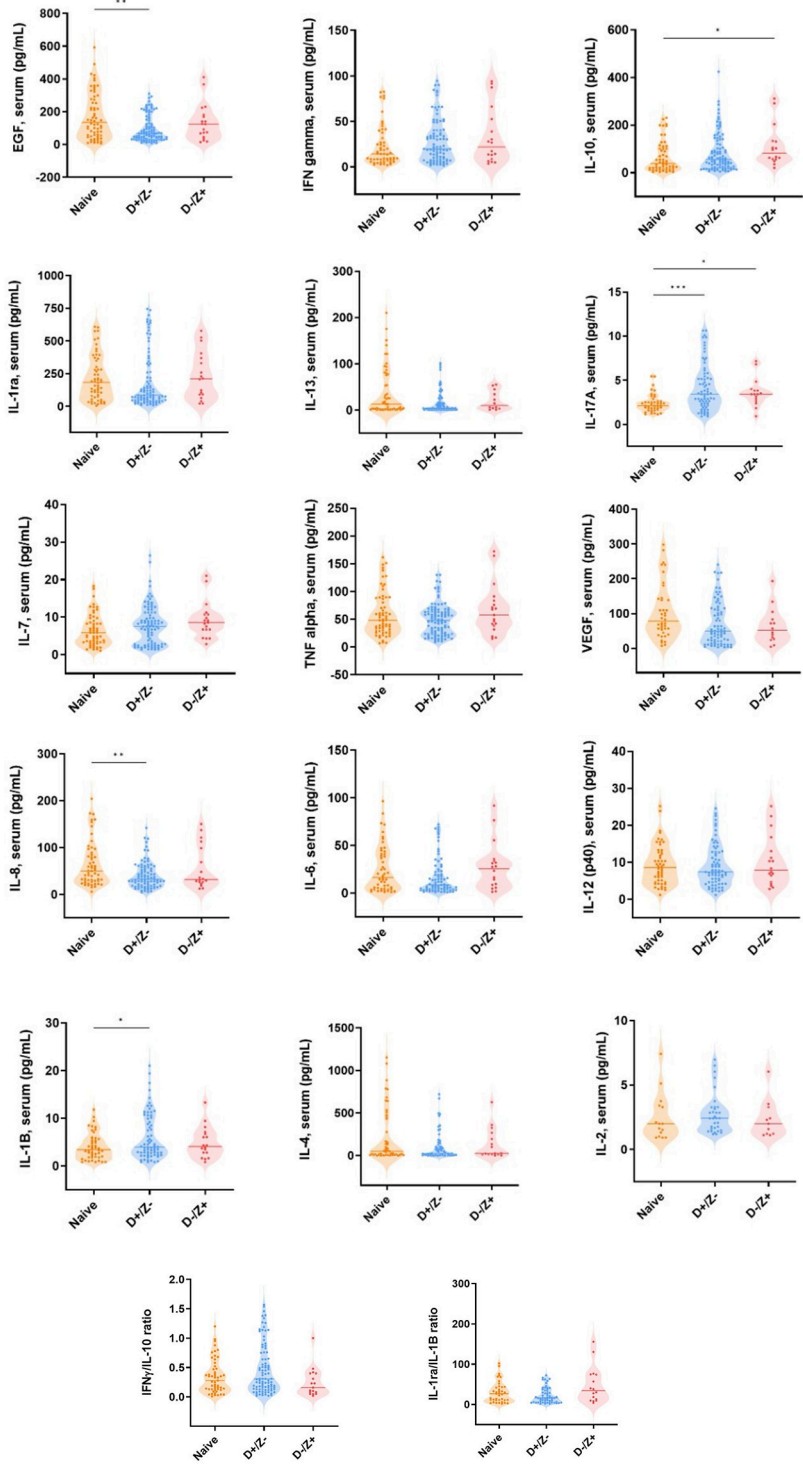

**Fig 5. Expression of cytokines according to serological sub cohorts.** * p < 0.05; ** p< 0.01.

decreased concentrations of epidermal growth factor (EGF), and IL-8 compared to dengue-naive individuals. The results from individuals with previous Zika infection (DV-ZV+) were comparable to those in previously dengue-naive individuals, except for IL-10 and IL-17A

(which were elevated in patients with previous Zika infection) (Fig 5). Notably, the IFN-γ, IL-10 and IL-1β:IL-1a ratios were not different in individuals with previous Zika infection, suggesting that no heightened immune or inflammatory responses occurred in this group.

## Discussion

Investigations have increasingly focused on understanding the mechanisms by which antibodies generated by a previous flavivirus infection might influence the evolution of subsequent infection by another flavivirus. In this study, we investigated whether previous Zika infection could influence the evolution of clinical outcomes of a subsequent dengue infection. Our observations suggest that previous ZIKV infection increased the risk of severe forms of dengue and hospitalizations, similar to what has been observed in secondary DENV infections. In contrast, our study suggests that the observed severity of dengue clinical outcomes don't seem to be influenced by ADE (increased viral load and anti-inflammatory cytokine levels), as prior infection to dengue was associated with a lower risk for development of severe forms of dengue disease. Conversely, patients with a previous Zika infection had neither higher viral titer nor cytokine profiles similar to ADE. For this reason, the mechanism involved in the clinical disease exacerbation by prior ZIKV immunity seem to differ from the classic ADE mechanism observed in secondary dengue infections (Fig 6).

As a way of overcoming the barrier of cross-reactivity between flaviviruses by conventional testing, we developed a peptide-based ELISA method to differentiate between previous infections of ZIKV and DENV. The results showed sensitivity of 95.9% and 95.6% to dengue and Zika, respectively. Such levels of sensitivity are similar or better than those reported for commercial tests [51, 52]. Our results demonstrated a significant differentiation potential. To reinforce this, the DV+/ZV+ group was excluded from all analyses in order to avoid dengue and

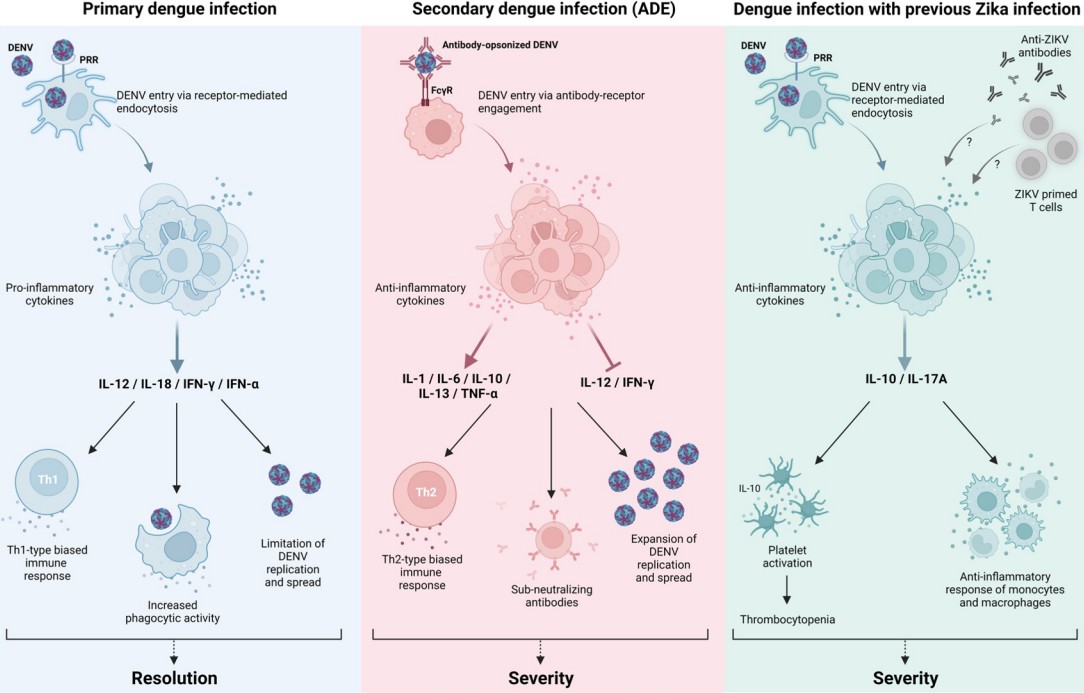

**Fig 6. Inflammatory response models in acute dengue infection in primary and secondary dengue infection [left and middle panel] and proposed mechanism of response after previous Zika infection [right panel]. Created with BioRender.com.**

Zika cross-reactivity in testing. Furthermore, we reasoned that the formation of immune complexes between NS1 and IgG might occur during acute phase of post primary infection and interfere with the test's sensitivity. Although, it does not affect the test's accuracy and its use in this study, once such immune complexes form, they represent part of the inflammatory response and free-NS1 IgG antibodies might be detected by the tests in the same way.

Considering the genomic similarity between the Zika and dengue viruses and the fact that ADE may occur among the different (genetically distinct but similar) dengue serotypes, it is reasonable to hypothesize that ZIKV infection could play a role in priming ADE in a subsequent DENV infection. Studies investigating the role of previous ZIKV infection on the outcomes of subsequent DENV infection, as represented here, are even more scarce [53]. Katzelnick et al. [34] found that Nicaraguan children with a history of Zika infection had a 12.1% probability of symptomatic DENV-2 infection in 2019/2020, while those with a history of dengue infection had a 9.2% probability and those with at least two dengue infections 2.9%, compared to a 3.5% probability in flavivirus-naïve children. It was also observed that intermediary anti-ZIKV antibody levels were associated with severe and symptomatic dengue infection, and that previous ZIKV infection modulated the future of dengue disease similar to prior DENV infection. Studies using animal models have suggested that previous ZIKV infection resulted in increased DENV viral load [54, 55], but also improved the immune response in terms of pro-inflammatory cytokine expression [54]. Another group, however, found that previous ZIKV infection did not affect DENV viremia or pro-inflammatory status in rhesus macaques [56]. Castanha et al. [57] demonstrated in a skin ex vivo model that previous ZIKV infection followed by DENV-2 infection resulted in increased density of DENV-2-infected cells and reduced amount of virus needed for myeloid cell infection. Similarly, prior ZIKV infection was also associated with greater severity of DENV infection in a mouse model [58, 59].

Our study showed that the introduction and circulation of ZIKV among the population of São José do Rio Preto in 2015–2016 induced immunity against the virus as assessed by the presence of ZIKV-specific IgG in patient sera. Preexisting ZIKV immunity may have to contribute to the occurrence of severe forms (DWS and SDD) of dengue, and higher hospitalization rates in 2019, as compared to dengue or Zika naive individuals. DENV viral titer were not increased and pro-inflammatory cytokine levels in plasma were not indicative of a cytokine storm, suggesting that the enhanced severity could not be attributed to ADE. However, we recognize that unpaired samples used for viral load analysis represent a limitation of this study. Paired samples collected during the acute phase might show different viral loads throughout the course of viremia, influenced by serotype, previous exposure to dengue, gender, and disease severity [60, 61].

The correlation between ADE and SDD has been described in the literature since the 1970s [62–65]. The pathogenic immune response observed in patients with ADE include higher serum levels of interleukin-2 (IL-2) and the soluble IL-2R receptor, soluble CD4, interferon-γ, (IFN-γ), interferon-α (IFN-α) (which remain elevated until convalescence), tumor necrosis factor-α (TNF-α), interleukin 1β (IL-1β), and platelet activation factor (PAF) [66]. Additionally, the interaction between platelets and monocytes in dengue infection triggers production of IL-1β, IL-8 and IL-10 [67]. The interplay of activated monocytes and CD4+ T cells results in the expression of IL-1β, IL-10, and IL-6 [58]. Reduced expression of IFN-γ and IL-12 in ADE may result in failure to induce DENV-specific neutralizing antibodies, which leads to increased viral growth and dissemination of infection [68]. Elevated plasma levels of EGF are also detected in patients with SDD [69].

It remains unclear why individuals previously infected with Zika experience more severe disease, and our results do not suggest that ADE and subsequent cytokine storm may be contributing factors. In the DV-/ZV+ group, which presented higher frequency of SDD cases or

hospitalization than the naïve group (DV-/ZV-), we observed higher levels of pro-inflammatory cytokines IL-10 and IL-17A. IL-10's pleiotropic effects in immunoregulation, inflammation are well documented [reviewed in [68]], while IL-17A modulates the proinflammatory response and has been associated to dengue severity and thrombocytopenia [70, 71]. IL-17A was the sole pro-inflammatory cytokine upregulated in DV-/ZV+ patients in comparison to the naïve group. Production of IL-17A is associated with activation of leukocytes and several other cell types, activation of effector mechanisms, tissue damage and multiple inflammatory diseases [72]. Thus, thrombocytopenia and development of SDD could be attributed to IL-17A production in DV-/ZV+ patients.

Notably, an important source of IL-17A is T CD4+ lymphocytes that differentiate in TH17 cells, specialized in IL-17A expression. Differentiation of TH17 cells is dependent T CD4+ lymphocyte TCR engagement and stimulation by several cytokines including IL-6 and IL-1β [73], which were detected in DV-/ZV+ patient sera. Although our data do not support the involvement of ADE in DV-/ZV+ patients, involvement of TH17 cells is probable and might raise the possibility of a pathogenic T cell memory response derived from original antigenic sin, as initially proposed by Rothman and colleagues for different serotypes of DENV [8, 74]. Previous ZIKV infection could lead to development of low-affinity cross-reactive immunological memory against epitopes in DENV, which would lead to pathogenic T helper cell responses in a subsequent infection, in this case, characterized by IL-17A production. Blockade of IL-17A function in inflammatory diseases such as psoriasis shows promise, but its application to the treatment of dengue or Zika remain unexplored [75]. In DV+/ZV-, we also observed higher levels of IL-17A compared to DV-/ZV-, but no evidence of thrombocytopenia. Such observations might be related to the complex cytokine interactions involved in dengue pathogenesis. Cytokine levels, such as INF-γ, IL-10, and TNF-α, did not differ between the DV+/ZV- and naïve groups and additional studies are needed to tease out their interactions and contribution to dengue pathogenesis.

Of note is that the average platelet count was lower in DV-/ZV+ patients, and leukopenia was more frequent in this than other groups. Thrombocytopenia in hospital admission was a predictor of hospitalization in logistic regression analyses, suggesting that aggravation of these clinical signs were key aspects of disease in this cohort. Platelets are key effectors of the immune response in viral infections, notably in dengue [67]. Platelet activation leads to secretion of multiple pro-inflammatory cytokines including IL-6, IL-8, TNF-α, CCL5 and IFNs, and also allows for activation of multiple cell types [67, 76]. Based on that, we cannot ignore that: (i) first, DV-/ZV+ presented lower platelet count, which might suggest platelet activation; (ii) second, this same serological group presented higher frequency of severe cases and hospitalization; (iii) third, DV-/ZV+ presented higher expression of IL-10 and IL-17A, responsible for the observed platelet activation; and (iv) fourth, thrombocytopenia was a risk predictor for dengue hospitalization (**Fig 6**). Altogether the evidence suggests that platelets may be involved in disease aggravation in DV-/ZV+ patients.

It is important to highlight that initially the DV+/ZV- group was designed to represent an ADE model in terms of cytokine expression. However, in this study, previous dengue infection was not associated with the anti-inflammatory cytokine profile observed in ADE, and these findings were not associated with more severe clinical course of the current dengue infection. In this study, the DV+/ZV- group did not necessarily comprise of cases of secondary dengue, but rather a report of previous dengue infection. This observation suggests the DV+/ZV- group comprises of multitypic DENV infections rather than the secondary infections commonly attributed to ADE, since Sao Jose do Rio Preto is hyperendemic for DENV circulation, where more than 70% of the city's population has been already infected with dengue [36],

confirming a reduced risk of disease already demonstrated in multitypic DENV infections elsewhere [28, 73, 74].

Other risk factors have been suggested as potential aggravating factors for the clinical forms of dengue, involved the association between age (children and the elderly) and progression to severe disease forms [77]. The older population was likely to have comorbidities for which infections likely required increased healthcare responses, including hospitalization. Furthermore, the management of these comorbidities also makes elderly patients a clinically vulnerable group. No statistically significant difference was found in the frequency of patients in each age group (p = 0.121) or the mean age between the groups (0.365).

In such a complex scenario of viral and host interactions and the successive emergence of arboviruses in favorable epidemiological conditions, it is imperative to link clinical practice to knowledge on the immunopathogenic mechanisms of disease, which are scarce. An interesting example of such approach is reported by Marzan-Rivera et al., who described that the dynamics of humoral immune response in rhesus macaques exposed to different flaviviruses can be modified by changing the sequence of flavivirus infection [78]. Additionally, understanding the characteristics that may be associated with unfavorable outcomes (such as hospitalization and development of SDD) remain a priority to improve the use of health care resources and management of patients.

This study was conducted in a tropical area where flaviviruses cocirculate, suggesting that previous Zika infection may be a risk factor for development of more severe forms of dengue disease and hospitalization. Contrary to the currently accepted mechanism of SDD in secondary dengue infection, the mechanism here appears not be associated with ADE, and neither DENV load, vascular leakage or cytokine storm were increased. In fact, individuals with previous Zika infection had increased levels of IL-10 and IL-17A and no difference in IFN-γ, IL-6, and IL-1β:IL-1ra ratios. These findings highlight the complex interactions between viruses and hosts and warn to fact of the knowledge acquired about the interaction among different dengue serotypes can not necessarily be applied to different flaviviruses. Thus, in addition to providing a diagnostic tool that seeks to mitigate the impact of cross-reactivity in assessing previous infection by dengue and Zika, this study, can provide a insights into the importance of comprehensive investigations on the influence of previous Zika immunity on subsequent dengue infections. Even though the mechanisms involved in triggering severe forms of dengue in DV-/ZV+ patients are not fully understood, studies like this one, suggesting potential pathways leading to severity, are important in order to inform the development of therapeutic and preventive countermeasures.

## Supporting information

**S1 Fig. Flow of sample selection for dengue-suspected cases with up to 7 days of symptoms onset.**
(TIF)

**S2 Fig. Differential diagnostic performance of the pepELISA and standard NS1 ELISA using validated samples.** Receiver Operating Characteristic (ROC) curves of pepELISA for the peptides (A) DV-15; (B) DV-20; (C) ZV-54; and standard ELISA against (D) DENV NS1 mix, and (E) ZIKV NS1. 'Control' identifies samples considered true negatives, and 'Patients' identifies samples considered true positives. Performance is demonstrated in True Positive Rate (Sensitivity %) versus False Positive Rate (100%—Specificity %).
(TIF)

**S3 Fig. Distribution of dengue-confirmed cases in São José do Rio Preto, São Paulo State, during 2019 epidemic, according to months and epidemiological weeks.**
(TIF)

**S1 Table. Characteristics of 1,043 dengue-confirmed cases eligible between December 2018 and November 2019, in São José do Rio Preto, São Paulo State.**
(XLSX)

**S2 Table. Measure of differential diagnostic performance for DENV NS1 protein, ZIKV NS1 protein, and for peptides DV-15, DV-20, and ZV-54.**
(XLSX)

**S1 Data. Dataset.** Deidentified data from samples included in the study.
(XLSX)

**S2 Data. Codebook.** Support information related to dataset.
(XLSX)

## Acknowledgments

We acknowledge colleagues from Hospital de Base de São José do Rio Preto and Municipal Health Secretary for support during sample and data collection. We also acknowledge Prof. Carlos Sariol for critical review of the manuscript.

## Author Contributions

**Conceptualization:** Cassia F. Estofolete, Lee Gehrke, Irene Bosch, Nikos Vasilakis, Maurício L. Nogueira.

**Data curation:** Cassia F. Estofolete, Maurício L. Nogueira.

**Formal analysis:** Cassia F. Estofolete, Alice F. Versiani, Maurício L. Nogueira.

**Funding acquisition:** Cassia F. Estofolete, Alice F. Versiani, Nikos Vasilakis, Maurício L. Nogueira.

**Investigation:** Cassia F. Estofolete, Alice F. Versiani, Fernanda S. Dourado, Bruno H. G. A. Milhim, Carolina C. Pacca, Gislaine C. D. Silva, Nathalia Zini, Barbara F. dos Santos, Flora A. Gandolfi, Natalia F. B. Mistrão, Pedro H. C. Garcia, Rodrigo S. Rocha, Flavio G. da Fonseca, Nikos Vasilakis.

**Methodology:** Cassia F. Estofolete, Alice F. Versiani, Nikos Vasilakis.

**Project administration:** Cassia F. Estofolete.

**Supervision:** Nikos Vasilakis, Maurício L. Nogueira.

**Validation:** Alice F. Versiani, Maurício L. Nogueira.

**Writing – original draft:** Cassia F. Estofolete, Alice F. Versiani, Lee Gehrke, Irene Bosch, Rafael E. Marques, Mauro M. Teixeira, Nikos Vasilakis.

**Writing – review & editing:** Cassia F. Estofolete, Nikos Vasilakis, Maurício L. Nogueira.

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
