## [Decision Letter · Decision Letter 0]

27 Jul 2023

Dear Prof. Nogueira,

Thank you very much for submitting your manuscript "Influence of previous Zika virus infection on acute dengue episode" for consideration at PLOS Neglected Tropical Diseases. As with all papers reviewed by the journal, your manuscript was reviewed by members of the editorial board and by several independent reviewers. In light of the reviews (below this email), we would like to invite the resubmission of a significantly-revised version that takes into account the reviewers' comments. 

Estofolete et al addressed in this study, comprising more than 1000 individuals, one of the most debated issues in the field of anti-flavivirus immune responses. If a previous infection(s) by a flavivirus can promote or reduce the risk of severe disease by a closely related flavivirus is critical for the management, treatment, and control (vaccines) of these flaviviruses. As expected, the review of this kind of work generates a combination of different comments from reviewers. 

We cannot make any decision about publication until we have seen the revised manuscript and your response to the reviewers' comments. Your revised manuscript is also likely to be sent to reviewers for further evaluation.

Sincerely,

Daniel Limonta, MD, PhD

Academic Editor

Abdallah Samy

Section Editor

Reviewer's Responses to Questions

**Key Review Criteria Required for Acceptance?**

**Methods**

-Are the objectives of the study clearly articulated with a clear testable hypothesis stated?

-Is the study design appropriate to address the stated objectives?

-Is the population clearly described and appropriate for the hypothesis being tested?

-Is the sample size sufficient to ensure adequate power to address the hypothesis being tested?

-Were correct statistical analysis used to support conclusions?

-Are there concerns about ethical or regulatory requirements being met?

Reviewer #1: Methods are generally appropriate.

Reviewer #2: acceptable

Reviewer #3: (No Response)

**Results**

-Does the analysis presented match the analysis plan?

-Are the results clearly and completely presented?

-Are the figures (Tables, Images) of sufficient quality for clarity?

Reviewer #1: Yes to all of the above.

Reviewer #2: yes

Reviewer #3: (No Response)

**Conclusions**

-Are the conclusions supported by the data presented?

-Are the limitations of analysis clearly described?

-Do the authors discuss how these data can be helpful to advance our understanding of the topic under study?

-Is public health relevance addressed?

Reviewer #1: Yes to all of the above.

Reviewer #2: yes

Reviewer #3: (No Response)

**Editorial and Data Presentation Modifications?**

Reviewer #1: N/A

Reviewer #2: accept

Reviewer #3: In this paper the authors looked at the flaviviruses’ (zika& dengue) immunity background in 1,043 laboratory confirmed dengue patients. The finding was that prior zika is a risk factor for severe dengue, even more than prior dengue. I find this observation extremely important as the entire world is occupied with the sequence of second dengue post dengue infection. If this observation is found to be true in subsequent studies then the concept of severe flavivirus infection should be revised to include prior other flavi or flavi viruses vaccines as a risk factor and not just various dengue serotypes.

The other interesting point is the importance that the authors gave to look for other explanations for severe disease rather than the common theory of ADE

The basis for the observation of tis study is the authors’ ability to identify accurately previous flaviviruses infection. They mentioned that they developed and validated a new peptide-based Enzyme-linked Immunosorbent Assay (ELISA) method for dengue and Zika detection, which I am not familiar with. I think they should elaborate more on this method, 

Minor comments

Abstract: 

Background: 

L. 38 "The simultaneous circulation of flaviviruses…" I prefer to say the co-circulation.

L. 39 "the hypothesis that immunity generated by a previous flavivirus could promote severe disease outcomes in subsequent infections by heterologous flavivirus" . The accepted theory until now is in regard to heterologous dengue serotypes and not among all flaviviruses.

Introduction

L. 70-" Infection by any of the four DENV serotypes causes acute febrile illness" –this is not true as most cases are probably asymptomatic.

L. 87 –" protocols in treating exacerbated disease outcomes" –the wording is not clear.

**Summary and General Comments**

Reviewer #1: Immune system interactions between closely-related flaviviruses during sequential infections can modulate an individual’s response to subsequent flavivirus infections. This is best exemplified in the setting of dengue virus, where pre-existing immunity to one dengue virus serotype can be protective, be neutral, or can enhance the risk of symptomatic disease during secondary infection, depending on the order in which those serotypes were encountered. In addition, emerging evidence from humans suggests that pre-existing immunity to Zika virus can enhance the risk of subsequent dengue virus infection. Here, Estofolete et al. use human samples collected from a region in Brazil that is hyperendemic for dengue virus but also previously experienced Zika virus circulation during the 2015-16 outbreak, to explore the role of pre-existing Zika virus immunity on subsequent dengue virus infection outcomes. They further evaluate the role that antibody-dependent enhancement might play in mediating the phenomena that they observe. Overall, this is a unique study with some unique aspects and thus an important contribution to the literature. However, I believe there are a few scientific issues and several things that need to be addressed to facilitate understanding by the reader. First, grammar and syntax need to be improved throughout, and the manuscript should be revised for clarity. In several instances it was difficult to follow exactly what was done with each sample. For example, a major component of this study is the development of an ELISA assay that can differentiate past ZIKV from DENV infections and vice versa, but it is not completely clear if this is simply proof-of-concept with a subset of samples or how this assay contributed in a significant way to delineating the 1,043 samples that were used for the analyses presented herein. 

In addition, more details should be presented on the dengue virus serotypes that individuals were infected with both as part of this study and in the past. If this information is not known that is okay, but some language should be included at the outset that describes how the analysis will focus on outcomes in a serotype-independent manner then. 

Other specific comments follow: 

Line 89: does it really matter if ZIKV emerges in areas where dengue is hyperendemic? If immune system interactions between DENV and ZIKV really matter, I would think that there is a concern about co-circulation of these two viruses wherever they are endemic. 

Line 91: the transition to this paragraph is awkward and confusing. 

Line 100-102: this should not be a stand alone paragraph and should be incorporated into the paragraph above describing problems associated with serological diagnosis of these two viruses. 

Line 103-120: this section of the introduction should be improved since it represents the underlying premise of the study. 

Line 304-305: I respectfully disagree with the statement that both DENV peptides efficiently separate DENV-positive from ZIKV-positive individuals. DV20 looks the best in this regard, but both DV20 and DV15 show some cross-reactivity with ZIKV-positive individuals—with DV15 showing significant cross-reactivity. All 3 peptides suffer from false negatives and false positives. In reality, it is probably unreasonable to expect these to be 100% and 100% specific but some context on relative performance to other assays would be appreciated here to understand how useful this tool is for the studies described here and for the field at large. 

Line 325: how were past DENV and ZIKV infections confirmed in the patient's medical records? Are these only individuals with a molecularly confirmed or NS1 antigen test? Or do these samples include individuals that had infections confirmed by serology? For those with past DENV infections, is it possible to know how many and the serotype?

Line 329-331: This section is a little confusing. It would be helpful to clarify that these specimens are from individuals in which the DENV infection could be their 1st, 2nd, or 3rd infection in a series of flavivirus exposures. 

Line 364: please add the sample size for this group with severe bleeding. 

Line 362-372: these analyses are for all dengue in general, but is there information on serotype, and if there is can you do more fine-scale analysis to understand serotype-specific interactions between DENV and ZIKV outcomes?

Line 370: I don’t understand the comparison listed here. The 2 outcomes used were resolution of infection and death? Death does not seem like an appropriate outcome to use for these types of analyses. I would suggest re-doing analyses with symptomatic infection and/or severe symptoms as the outcome. 

Discussion: the discussion is overly long and could be tightened up to focus on how this study is or is not consistent with the existing literature, potential mechanisms that might underlie the interaction, and where the field might go from here.

Reviewer #2: Novel study showing the casual relation of zika and Dengue sevirity,but need to find evidence of antibody-dependent enhancement can be emphasised

Reviewer #3: The basis for the observation of tis study is the authors’ ability to identify accurately previous flaviviruses infection. They mentioned that they developed and validated a new peptide-based Enzyme-linked Immunosorbent Assay (ELISA) method for dengue and Zika detection, which I am not familiar with. I think they should elaborate more on this method,

PLOS authors have the option to publish the peer review history of their article (what does this mean?). If published, this will include your full peer review and any attached files.

Reviewer #1: No

Reviewer #2: No

Reviewer #3: No
---

## [Editor Report · Decision Letter 1]

5 Oct 2023

Dear Prof. Nogueira,

We are pleased to inform you that your manuscript 'Influence of previous Zika virus infection on acute dengue episode' has been provisionally accepted for publication in PLOS Neglected Tropical Diseases.The authors addressed carefully all the comments and observations of the three reviewers. As expected, such a conundrum like the immune responses in sequential flaviviral infections will raise disagreements between the presented results and previous studies. Perhaps, because of this, the discussion section was kept longer than usual. Furthermore, often, many field studies in hyperendemic areas of arboviruses do not count on accurate measurements of a specific aspect of the antiviral antibody response due to a number of logistical reasons. However, in many instances, the significance of the results in the studied human subjects overcomes a minor detail in the lab data collection.

Best regards,

Daniel Limonta, MD, PhD

Academic Editor

Abdallah Samy

Section Editor

---

## [Editor Report · Acceptance letter]

24 Oct 2023

Dear Prof. Nogueira,

We are delighted to inform you that your manuscript, "Influence of previous Zika virus infection on acute dengue episode," has been formally accepted for publication in PLOS Neglected Tropical Diseases.

Best regards,

Shaden Kamhawi

co-Editor-in-Chief

Paul Brindley

co-Editor-in-Chief
